# Insomnia, Perceived Stress, and Family Support among Nursing Staff during the Pandemic Crisis

**DOI:** 10.3390/healthcare8040434

**Published:** 2020-10-26

**Authors:** Athanasios Tselebis, Dimitra Lekka, Christos Sikaras, Effrosyni Tsomaka, Athanasios Tassopoulos, Ioannis Ilias, Dionisios Bratis, Argyro Pachi

**Affiliations:** 1Psychiatric Department, Sotiria Thoracic Diseases Hospital of Athens, 11527 Athens, Greece; psyxdoc@sotiria.gr (A.T.); lekkadim@yahoo.gr (D.L.); tsomaka@gmail.com (E.T.); blatomgr@hotmail.com (A.T.); dionbratis@yahoo.gr (D.B.); irapah67@otenet.gr (A.P.); 2Nursing Department, Sotiria Thoracic Diseases Hospital of Athens, 11527 Athens, Greece; cris.sikaras@gmail.com; 3Department of Endocrinology, Maternity Hospital “Helena Venizelou”, 11521 Athens, Greece

**Keywords:** insomnia, stress, family support, nurses, COVID-19

## Abstract

The COVID-19 pandemic is likely to cause mental health issues, especially for healthcare professionals. The aim of this study was to investigate levels of perceived stress, insomnia, and the sense of family support among nurses in pandemic conditions. We administered in a sample of 150 nurses from different hospital departments during the COVID-19 pandemic the Athens Insomnia Scale (AIS), Perceived Stress Scale (PSS), and Family Support Scale (FSS). Individual and demographic data were recorded. In total, 120 women and 30 men completed the study questionnaires. Almost half of the participants (49.7%) reported the presence of insomnia and more than half (50.3%) experienced increased stress levels. Scores on the Athens Insomnia Scale correlated positively with Perceived Stress Scale scores (*p* < 0.01), and negatively with Family Support Scale scores (*p* < 0.01). Significantly negative correlations were observed among scores on the Perceived Stress Scale and the Family Support Scale (*p* < 0.01). The regression models revealed that ‘scores on Perceived Stress Scale’ and ‘years of work experience’ were significant predictors of ‘scores on Athens Insomnia Scale’, each explaining 43.6% and 2.3% of the variance. ‘Scores on Athens Insomnia Scale’ and ‘scores on Family Support Scale’ were significant predictors of ‘scores on Perceived Stress Scale’, explaining 43.7% and 9.2% of the variance. In conclusion, we confirmed that working with COVID-19 patients has a negative impact on the sleep of nurses, possibly mediated by increased levels of stress. Family support, as a protective factor, appears to moderate the deleterious consequences of stress.

## 1. Introduction

Since December 2019, the whole world has been experiencing an unprecedented situation due to the emergence of severe acute respiratory syndrome coronavirus 2 (SARS-CoV-2), causing the pandemic of novel coronavirus disease (COVID-19). Presently, frontline professionals working in designated hospitals for COVID-19 treatment experience a sense of intense psychological pressure [1]. High risk of infection, inadequate protection from contamination, increased workload, discrimination, isolation, and uncertainty about the outcomes of the pandemic are all expected to impose stress upon healthcare workers, seriously affecting mental health, including sleep hygiene [2].

Sleep is a biological requirement for human life, alongside food, water, and air, and is vital for maintaining life and health as well as safe working conditions. Seven to eight hours sleep a night is associated with a lower risk of obesity, diabetes, hypertension, myocardial infarction, and stroke [3], and a reduced risk of injuries and fatigue-related errors [4]. Medical and nursing staff are usually exposed to shifts and long working hours. These demanding occupational programs contribute to employee sleep disorders [5].

According to previous studies on SARS and Ebola epidemics, the onset of a sudden life-threatening disease could lead to extraordinary amounts of pressure on healthcare professionals [6,7]. High-intensity work, physical and emotional exhaustion, inadequate personal protection equipment, risk of nosocomial infection during hospitalization, helplessness, fear, anxiety and concern for patients and family members, and the need to make sense of the morally challenging decisions being made can have dramatic effects on their physical and mental well-being [2]. Their resilience can be further compromised by isolation and loss of social support, risk of contagion and infection of friends and relatives, as well as drastic, often annoying changes in working conditions. Previous studies have reported that healthcare staff, especially those who work in the frontline during viral epidemic outbreaks, are at high risk of developing mental health issues, including stress, anxiety, depressive symptoms, anger, insomnia, fear, and sleep disorders [8,9,10].

As the COVID-19 pandemic took hold, nurses were on the front line of health and social care in the most extreme of circumstances. In similar outbreaks, nurses have had the highest levels of occupational stress and resulting distress compared to other health personnel [11,12,13]. Having to enter the negative pressure ward to care for patients after only undergoing a brief training, working in the intensive care units and the infection departments, spending hours each day putting on and removing airtight protective equipment, being transferred to other non-anti-epidemic positions, and in the meanwhile worrying about becoming infected or infecting family members at times may compromise safety and wellbeing and can lead to adverse mental health outcomes [14].

Family support refers to the sense of perceived support from the family environment [15]. It is an important element of social support. Evidence on the influence of family support on chronic illness self-management behaviors shows that perceived family support is positively associated with improved outcomes among individuals with diabetes mellitus [16,17], chronic obstructive pulmonary disease [18], and even in lung cancer patients [19]. Additionally, research highlights the negative correlation between family support and depression [20], while another study reported a negative correlation between family support and burnout [21]. The role of family support is likely to become more influential in situations where entire societies are placed under tightened quarantine restrictions.

The aim of the study was to investigate the prevalence of insomnia and perceived stress and evaluate their possible correlation with the sense of family support in a specialized COVID-19 hospital’s nursing staff during the pandemic crisis.

## 2. Subjects and Methods

### 2.1. Research Design

This cross-sectional study was conducted in one of the largest hospitals in Greece after approval from the Clinical Research Ethics Committee (Number 12253/7-5-20). Participants gave their written consent by responding to the questionnaire. The hospital received the first suspected COVID-19 case in February 2020 and remained a specialized hospital treating exclusively COVID-19 patients until May 2020. The study took place in the second half of May 2020.

### 2.2. Study Participants

In total, 150 nurses (out of 679) were randomly selected, and agreed to participate anonymously by completing self-reported questionnaires. Participants worked in departments of respiratory medicine, fever clinics, intensive care units, or in other non-anti-epidemic positions, during the past month.

### 2.3. Measurement Tools

Demographic and social data from the study participants included age, gender, education, and marital status. Professional and work information included their title, role, department that they worked in during the last month, and work experience.

### 2.4. Athens Insomnia Scale (AIS)

The severity of sleep disturbances was measured by the Athens Insomnia Scale (AIS) [22]. The scale is a self-assessment psychometric instrument designed for quantifying sleep difficulty based on the International Classification of Diseases 10th revision (ICD-10) criteria for insomnia. It consists of eight items: the first five pertain to sleep induction, awakenings during the night, final awakening, total sleep duration, and sleep quality; while the last three refer to well-being, functioning capacity, and sleepiness during the day. In this scale, the sleep difficulty severity is measured based on a 4-point Likert scale, since last month. The scores ranged from 0 (meaning not being a problem) to 3 (meaning more acute sleep difficulties). A cut-off score of 6 has been determined to distinguish between insomniacs and healthy subjects [23]. The validated Greek version of the scale was employed. The internal consistency (Cronbach’s alpha) for the total scale was 0.89 [22].

### 2.5. Perceived Stress Scale (PSS)

PSS was used to evaluate the perception of stressful experiences. The 10-item PSS [24] measures global perceived stress experienced, by asking the respondent to rate the frequency of his/her feelings and thoughts related to events and situations that occurred across the past 30 days, on a 5-point scale (0 = never, 1 = almost never, 2 = once in a while, 3 = often, 4 = very often). Total scores range from 0 to 40, with higher scores indicating higher perceived stress. Scores ranging from 0–13 would be considered low stress, from 14–26 moderate stress, and from 27–40 high perceived stress. As the PSS is not a diagnostic instrument, there are no cut-off scores. The validated Greek version of the scale was utilized. The internal consistency, as indicated by Cronbach’s alpha, for the total scale was 0.82 [24].

### 2.6. Family Support Scale (FSS)

To evaluate perception of family support, we used the family support scale [15], which aims to record the sense of support that a subject receives from the members of his/her family (with whom he/she lives). The scale consists of 13 items, which are answered on a Likert scale, ranging from 1 (“I disagree a lot”) to 5 (“I agree a lot”). The scale is self-administered and all of the items focus on the interrelations of individuals that live together. High scores correspond to an increased sense of family support. Individuals that live alone did not complete the scale. The validated Greek version of the scale was utilized. The internal consistency (Cronbach’s alpha) for the total scale was 0.82 [15].

### 2.7. Statistical Analysis

Statistical analysis was done with SPSS v.20 (IBM Corp, Armonk, NY, USA). All variables were assessed with the use of descriptive statistics and values were expressed as the mean ± standard deviation for continuous variables. The prevalence of insomnia and perceived stress levels were determined in percentages. All distributions of the continuous variables were normal (one—sample Kolmogorov–Smirnov test, *p* > 0.05). The independent groups *t*-test was performed to evaluate differences on continuous variables as to gender. The one-sample *t*-test was employed to determine differences on scores between nursing staff and the Greek general population. One-way Analysis of variance (ANOVA) was used to test for differences on scores among included participants from various hospital departments. Pearson’s correlation and stepwise linear regression analysis were also used for the evaluation of continuous variables. Statistical significance was set at *p* < 0.05 (two-tailed).

## 3. Results

A total of 150 nurses (120 women and 30 men) completed the study questionnaires. Demographic characteristics and work experience of the study participants are shown in Table 1.

No statistically significant difference was noted between men and women regarding age and years of work experience (independent *t*-test, *p* > 0.05). Additionally, no statistically significant gender differences were observed in mean scores on the Athens Insomnia Scale, Perceived Stress Scale, and Family Support Scale (Table 1, *p* > 0.05).

In addition, the study did not find significant differences in scores on the Athens Insomnia Scale, Perceived Stress Scale, and Family Support Scale among participants who worked in fever clinics or wards for patients with COVID-19 and nurses who worked in non-frontline departments, possibly owing to the effect of the periodical rotation of personnel (ANOVA post hoc, *p* > 0.05). In total, 49.7% of participants scored above cutoff on the Athens Insomnia Scale, indicating the presence of sleep disorders. On the Perceived Stress Scale, 49.7% of responders expressed low levels of stress, 45.5% moderate stress, and 4.8% high perceived stress.

Males stated an increased sense of family support with a mean value of 54.5 on the Family Support Scale, a result that differs significantly from the reference value of 50.8 [15] in the standardized version of the scale in a sample of Greek healthcare professionals (one sample t test *p* < 0.05). Similarly, the mean Family Support Scale value in female participants was significantly different from the corresponding reference value of 47 [15] in the standardized version of the scale (52.16 versus 47, one sample t test *p* < 0.01).

Scores on the Athens Insomnia Scale correlated positively with increasing age and years of work experience (*p* < 0.05, Table 2).

There were significant positive correlations among Athens Insomnia Scale scores and Perceived Stress Scale scores (*p* < 0.01), and negative correlations among Athens Insomnia Scale scores and Family Support Scale scores (*p* < 0.01). Additionally, significantly negative correlations were observed among Perceived Stress Scale scores and Family Support Scale scores (*p* < 0.01, Table 2).

Before performing stepwise multiple regression, we checked all the variables for the assumption of homoscedasticity (with inspection of scatterplots), and for absence of multicollinearity (with a variance inflation factor < 5). Stepwise multiple regression analysis was then conducted to identify the best predictors of the dependent variable ‘scores on Athens Insomnia Scale’ among the independent variables that correlated significantly with the dependent variable in the correlation analysis (scores on Perceived Stress Scale, scores on Family Support Scale, age, and years of work experience) and to examine their contribution to the variation (expressed as R^2^) in the dependent variable. The final regression model showed that from all variables entered into the equation, ‘scores on Perceived Stress Scale’ and ‘years of work experience’ were significant predictors of ‘scores on the ’Athens Insomnia Scale’, each explaining 43.6% and 2.3% of the variance (F_2,115_ = 48.897, *p* = 0.000, Table 3).

On the basis of the results of the bivariate analyses, a stepwise multiple regression test was performed to identify the best predictors of the Perceived Stress Scale from the independent variables that correlated significantly with it in the correlation analysis (Family Support Scale and Athens Insomnia Scale scores) and to examine their contribution to this dependent variable’s variation (expressed as R square). The final regression model showed that scores on the Athens Insomnia Scale and scores on the Family Support Scale were significant predictors of the dependent variable, explaining 43.7% and 9.2% of the variance, respectively (F_1,116_ = 65.040, *p* = 0.000, Table 4), whereas age and work experience were not significant (*p* > 0.05, Table 4).

## 4. Discussion

This cross-sectional study suggested a high prevalence of sleep disorders among nurses working in a specialized COVID-19 hospital, in the past month, during the pandemic. A recent study in the Greek population, using the Athens Insomnia Scale to explore sleep difficulties during the COVID-19 pandemic, reported sleep problems in 36.7% of the general population [25]. A survey of insomnia among medical staff members in China found that more than one-third of the medical staff suffered insomnia symptoms during the COVID-19 outbreak [26]. In another study [27], scores on the Athens Insomnia Scale to evaluate sleep disturbances of Chinese frontline medical workers were not statistically different from the findings of our study (6.3 ± 4.2 versus 5.95 ± 4.238, *p* > 0.05). Yet, even before the pandemic crisis, insomnia seems to be prevalent among nurses in public hospitals of Greece [28] and in other countries, recently [29] and in the past [30].

More than half of the nursing staff who responded to questionnaires expressed moderate to high levels of stress. The nursing profession is widely acknowledged to be one of the most highly stressful occupations [31]. The level of stress among nurses generally ranges from moderate to high [32,33]. Literature suggests that nursing staff are psychologically more vulnerable to the pressure caused by the pandemic [34]. These results may be partly biased because of the gender effect [35], taking into account that the majority of nurses are females, but could also be attributed to the fact that they may face extreme challenges. A higher risk of exposure to patients with COVID-19 as they spend more time in wards providing nursing care to patients and being responsible for the collection of sputum for the detection of viruses or working continuously and intensively in the isolation wards justify the highest levels of occupational stress and resulting distress compared with other health personnel. In addition, due to their intimate contact with patients, they may be more exposed to moral injuries associated with suffering, death, and ethic dilemmas [36].

Results from the linear regression analysis in this study suggested that the effect of the coronavirus outbreak on sleep difficulties has partly been mediated by stress. Research indicates that stress is closely related to sleep quality. The Hypothalamic Pituitary Adrenal axis (HPA) axis, central catecholamine systems, and sympathetic system play an important role in the regulation of the sleep–wake cycle and there is good evidence that dysregulation of the neural and neuroendocrine mediators of the stress response can lead to sleep disturbances [37]. In turn, sleep disorders may appear to lead to further HPA axis dysregulation, thereby promoting a vicious cycle of stress and insomnia [38]. From another point of view, there is evidence of a bidirectional association between insomnia and loneliness [39], providing a different approach for the positive influence of family support.

The role of perceived family support seems to be important in moderating stress levels possibly by reducing the perception of the threat of stressful events and the physiological response and inappropriate behavior that can result from stress, leading to improved self-efficacy and a sense of professional achievement and acting protectively against the development of burnout [40]. Additionally, an increased sense of family support, reported by our study, in the midst of a pandemic, may be related to the recognition of the absolute need and the significant contribution of the nursing profession [41,42,43].

The findings from this study showed that family support of the nursing staff did not directly influence rates of insomnia but followed an indirect pathway via stress. In more detail, family support has a protective role on nurses’ sleep by having a positive contribution to reducing stress levels; however, if this increased sense of family support is temporary, it is not sufficient. In this context, it is highly recommended that nurse managers and policy makers pay more attention to this phenomenon. Early targeted interventions should be considered that will include a network of specialized mental healthcare services to address psychological care needs and attenuate the possibility of escalating complications. Hospital administrators should be aware of the extent and sources of stress and psychological distress among frontline healthcare workers during disease outbreaks and implement interventions at the management level that will alleviate the impact of psychological pressure.

The message from the World Health Organization for healthcare workers forewarned that managing mental health and psychosocial wellbeing during this pandemic is as important as managing physical health [44]. In other words, even before the crisis begins to recede, mental health issues need to be addressed in order to avoid long-lasting impact on healthcare workers.

## 5. Conclusions

High rates of sleep disturbances were observed among nurse participants during the pandemic period. Reported stress levels correlate significantly with sleep disorders. Sense of family support is negatively associated with perceived stress and indirectly with insomnia. Lessons learned from the pandemic call for countermeasures and other recommended safety practices in order to protect and care for healthcare staff who find themselves unexpectedly on the dangerous front lines of disease response.

## Figures and Tables

**Table 1 healthcare-08-00434-t001:** General characteristics of nursing staff and scores on Athens Insomnia Scale (AIS), Perceived Stress Scale (PSS), and Family Support Scale (FSS) as to gender.

Subjects	Descriptive Statistics	Age	Work Experience (in Years)	AIS	PSS	FSS
Men N = 30	Mean	44.27	16.77	5.20	13.57	54.50
	SD	11.01	12.088	3.79	7.82	8.02
Women N = 120	Mean	41.79	17.36	6.18	14.91	52.16
	SD	10.65	11.167	4.34	6.89	7.95
Total N = 150	Mean	42.29	17.24	5.98	14.64	52.67
	SD	10.73	11.32	4.24	7.09	7.99

**Table 2 healthcare-08-00434-t002:** Correlations among continuous variables: age, work experience (in years), Athens Insomnia Scale (AIS), Perceived Stress Scale (PSS), and Family Support Scale (FSS).

Variables	Age	Work Experience (in Years)	AIS	PSS
Work experience (in years)	0.925 **			
Athens Insomnia Scale (AIS)	0.186 *	0.204 *		
Perceived Stress Scale (PSS)	0.106	0.115	0.650 **	
Family Support Scale (FSS)	−0.029	−0.109	−0.388 **	−0.536 **

* *p* < 0.05 or ** *p* < 0.01.

**Table 3 healthcare-08-00434-t003:** Stepwise multiple regression analysis of factors predicting insomnia.

Dependent Variable: Athens Insomnia Scale (AIS)	Beta	*t*	*p*
Age (in years)	−0.010	0.056	0.956
Work experience (in years)	0.153	2.221	0.028 *
Family Support Scale (FSS)	−0.038	−0.461	0.645
Perceived Stress Scale (PSS)	0.647	9.394	0.000 **
R Square	0.460		
Durbin-Watson	1.976		
F_2,115_	48.897		
Significance	0.000 **		
Variance inflation factor	1.008		

Beta = Standardized Regression Coefficient; Correlations are statistically significant at the * *p* < 0.05; or ** *p* < 0.01 level.

**Table 4 healthcare-08-00434-t004:** Stepwise multiple regression analysis of factors predicting stress.

Dependent Variable: Perceived Stress Scale (PSS)	Beta	*t*	*p*
Age (in years)	−0.028	−0.431	0.667
Work experience (in years)	−0.060	−0.916	0.362
Family Support Scale (FSS)	−0.329	−4.753	0.000 **
Athens Insomnia Scale (AIS)	0.533	7.708	0.000 **
R Square	0.529		
Durbin-Watson	2.022		
F_1,116_	65.040		
Significance	0.000 **		
Variance inflation factor	1.181		

Beta = Standardized Regression Coefficient; Correlations are statistically significant at the ** *p* < 0.01 level.

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
