# Peer review of "Insomnia, Perceived Stress, and Family Support among Nursing Staff during the Pandemic Crisis"

_healthcare, 2020, doi:10.3390/healthcare8040434_

Round 1

Reviewer 1 Report

This is a timely work. Its brevity and simplicity is commendable.

However, I wish there could be a recommendation as part of the concluding paragraph. The conclusion is obvious - health care workers need more supports. How do we help them get this support? Let these recommendations be made clearer in your conclusion.

Also note the following minor issues:

Section 3.1, Line 32:

"Since December 2019, the whole world has been experiencing …" may be considered.

"Section 2.2, Line 87:

"during the past month" may be considered.

Section 2.3 - Line 91, "worked in during last month, and work experience."

Section 2.4 - Line 95, "on the ICD-10 criteria." What is ICD-10? Please define ICD.

In Table 4 you mentioned FS could that be FSS? Also on Line 167 corresponding to the table is "Correlations" the best word there? These are just for your considerations nothing serious.

Section 2.7, Line 123 - 129 seems to be comments that should not be part of the paper - "Materials and Methods …. appropriately cited".

Author Response

1st Reviewer

[1]. I wish there could be a recommendation as part of the concluding paragraph. The conclusion is obvious - health care workers need more supports. How do we help them get this support? Let these recommendations be made clearer in your conclusion.

Following the first reviewer’s suggestions concerning the need of clearer recommendations as part of the concluding paragraph, we enriched the discussion and conclusion and included specific measures and directions that can be taken, in order to help nurses get the support they need.

[2]. Regarding the following minor issues:

Section 3.1, Line 32:

"Since December 2019, the whole world has been experiencing …" may be considered.

"Section 2.2, Line 87:

"during the past month" may be considered.

Section 2.3 - Line 91, "worked in during last month, and work experience."

We have addressed all above issues, following the reviewer’s suggestions.

[3]. Section 2.4 - Line 95, "on the ICD-10 criteria." What is ICD-10? Please define ICD.

ICD-10 refers to the International Classification of Diseases (10th revision; ICD-10). This has been clarified in the revised version of the manuscript.

[4]. In Table 4 you mentioned FS could that be FSS? Also on Line 167 corresponding to the table is "Correlations" the best word there? These are just for your considerations nothing serious.

The reviewer is right, it is indeed FSS. This has been addressed in the revised version of the manuscript. Furthermore “Correlations” has been removed from the Table’s footer.

[5]. Section 2.7, Line 123 - 129 seems to be comments that should not be part of the paper - "Materials and Methods …. appropriately cited".

This was an irrelevant intrusion which has been deleted from the revised version paper. Again, we want to thank our reviewer for the constructive comments and the specific suggestions. 

Reviewer 2 Report

This manuscript describes a very important topic, and although the sample size is low, the sample is randomly selected which is a strength, it is a cross-sectional study which makes causality impossible to predict, however, the topic is of so high interest that it is also need for cross-sectional studies in this field right now. In particular I find it is interesting that family support is related to stress, but not to insomnia in the sample. Below are some specific comments and recommendations to the manuscript:

Abstract: Would prefer names over abbreviations (eg for PSS), it is hard to read. Consider this for the rest of the manuscript as well. 

Introduction, page 1, line 35-38: What in this sentence is from reference (1)? As far as I can see the article in reference 1 does not report on any results concerning a relationship between increased workload, discrimination ect. and mental health/sleep. Rather, it is about the mental health status of medical personell during the pandemic. 

Line 39-41 starts with “Studies suggest”, but does not include a reference. Which studies suggest this? And what is “sufficient sleep quality”? Maybe consider rephrasing it to just “Sleep is a biological requirement for human life, alongside (..)”, as this particular statement about sleep is quite certain, and not just something some “studies suggest”. 

Next sentence, line 41: 7-8 hours of sleep - which studies is this from? And lower risk compared to what? Nine hours? Six hours? 

Line 45: “According to previous studies” - which studies? Where is the reference? Reference should come the first time the study is mentioned. In general, the references need to be distributed to the right sentences in this whole section. Each sentence here seem to need a reference. 

Line 58: “compared with other groups” which other groups? Health personnel? The community in general? 

Line 64 - where is this definition on family support from? 

Aim, line 73: Parts of the aim is to investigate an “effect”, but since this is a cross-sectional study, you can not investigate effect, this should be re-written. 

One aim is to investigate sleep disorders, but only insomnia is measured. Why? Is it correct to name this “sleep disorders” then? Or is the insomnia scale used a measure of sleep disorders in general? The authors may explain why they have chosen to measure insomnia and not other sleep disorders, or frame it differently. 

Methods: Are the instruments used validated in previous research? Which language are they used in and are these versions validated? What are the Cronbach alphas for the scales? 

The AIS description, line 94-95: The authors write that the instrument is designed for quantifying sleep difficulty based on the ICD-10 criteria. The ICD-10 criteria for what? Insomnia? 

Line 123-129: Is this a text about the requirements of the methods section in the journal? Should be removed? 

It would also be nice with a little more information about how the statistical analyses were performed, what specific part of the study aims was investigated with for example the t-test, and what with the regression? 

Results

There are very many tables, and too much focus on gender in the beginning. This is not part of the study aims, why are there separat analyses looking at this? I would suggest considering collapsing table 1, 2 and 3 into one where you show Mean and SD in the first column after the variables (for non-categorical variables, show frequencies for categorical variables as gender), then show the correlations - remove the n and the sig rows that now are in the correlation table. In that way it is easier to read the correlations. 

line 132: What is “working data”?

Part of the aim is to look at prevalence of sleep problems. Do you have any data from other samples/the same/similar samples at a time when there is no pandemic, to compare the % with? That 4.8% of the participants reported high levels of perceived stress does not give much information without anything to compare it with. A prevalence of 49.7% for insomnia sounds high, but how is this compared to before COVID-19, or compared to others? This is mentioned in the beginning of the discussion, but would it be possible to get these data from the general population or other studies to analyse the differences? 

Correlation analysis: It would be easier for the reader to read this table if the dependent variables (in the regression analysis) were presented at the bottom of the table, and that all the tables had the same order (demographic variables first, then FS, then PSS/AIS)

The regression analysis: I am not surprised that family support was not significantly related to AIS when controlling for PSS. The correlation between PSS and AIS is high. Were all the assumptions for performing a regression analysis met in the data? 

It is interesting that family support is related to stress, but not to insomnia. It would be interesting to see if the results changed if the Stress analysis included the same variables as the Insomnia analysis (that is, also including age in the analysis for stress even though it was not correlated to stress in the correlation analysis - there are different opinions as to including variables with low/no correlation in a regression or not).

line 170 “the independent variables that showed significant associations” - is this in the correlation analysis?

Discussion: 

Line 189: “Literature suggests” What literature? What is the reference? 

There need to be a higher awareness of the causality of the variables (or lack of due to the cross-sectional design) in the discussion in general, especially when discussing the results from the regression. 

line 204, sentence reads: “there is evidence of a bidirectional association between insomnia and loneliness [28], providing an explanation for the positive effects of family support.” Which effects of family support are the authors talking about here? In the regression family support was not significantly related to insomnia. This is also relevant for the last sentence in the conclusion. 

The implications of the main findings could be communicated more clearly. What is the most important finding and how can this improve the sleep/work of nurses during the pandemic? I think it is interesting that family support is related to stress, but not to insomnia. Why is this? 

Author Response

2nd Reviewer

[1]. Abstract: Would prefer names over abbreviations (eg for PSS), it is hard to read. Consider this for the rest of the manuscript as well. 

We replaced all abbreviations with complete names throughout the text.

[2]. Introduction, page 1, line 35-38: What in this sentence is from reference (1)? As far as I can see the article in reference 1 does not report on any results concerning a relationship between increased workload, discrimination ect. and mental health/sleep. Rather, it is about the mental health status of medical personell during the pandemic. 

In the revised version of the manuscript we allocated the references appropriately to the right sentences.

[3]. Line 39-41 starts with “Studies suggest”, but does not include a reference. Which studies suggest this? And what is “sufficient sleep quality”? Maybe consider rephrasing it to just “Sleep is a biological requirement for human life, alongside (..)”, as this particular statement about sleep is quite certain, and not just something some “studies suggest”. 

We appreciate the suggestion. The reviewer is right, there is no need for references for such common statements. The relevant text has been modified as follows: “Sleep is a biological requirement for human life, alongside food, water, and air and is vital for maintaining life and health as well as safe working conditions.”

[4]. Next sentence, line 41: 7-8 hours of sleep - which studies is this from? And lower risk compared to what? Nine hours? Six hours? 

In the revised version of the manuscript we included the following references that support our statement:

  1. Chaput JP, McNeil J,  Després JP, et al. Seven to Eight Hours of Sleep a Night Is Associated with a Lower Prevalence of the Metabolic Syndrome and Reduced Overall Cardiometabolic Risk in Adults. PLoS ONE 2013 Sep 5;8(9):e72832.
  2. Rogers AE. The Effects of Fatigue and Sleepiness on Nurse Performance and Patient Safety. In: Hughes RG, editor. Patient Safety and Quality: An Evidence-Based Handbook for Nurses. Rockville (MD): Agency for Healthcare Research and Quality (US); 2008 Apr.

[5]. Line 45: “According to previous studies” - which studies? Where is the reference? Reference should come the first time the study is mentioned. In general, the references need to be distributed to the right sentences in this whole section. Each sentence here seem to need a reference. 

Following the reviewer’s suggestion, in the revised version of the manuscript we allocated the references appropriately to the right sentences.

[6]. Line 58: “compared with other groups” which other groups? Health personnel? The community in general? 

In the revised version of the manuscript we added the following: ‘…compared to other health personnel…’ [as evidenced in the cited references].

[7]. Line 64 - where is this definition on family support from? 

The definition on family support is from following published report:

15.Tselebis A, Anagnostopoulou T, Bratis D, Moulou A., Maria A., Sikaras C., Ilias I., Karkanias A., Moussas G., Tzanakis N. The 13 item Family Support Scale: Reliability and validity of the Greek translation in a sample of Greek health care professionals. Asia Pacific Family Medicine 2011, 10, 3.

[8]. Aim, line 73: Parts of the aim is to investigate an “effect”, but since this is a cross-sectional study, you cannot investigate effect, this should be re-written. 

In the revised version of the manuscript we rephrased the aim of the study as follows: “The aim of the study is to investigate the prevalence of insomnia and perceived stress and evaluate their possible correlation with the sense of family support in a specialized COVID-19 hospital’s nursing staff during the pandemic crisis”.

[9]. One aim is to investigate sleep disorders, but only insomnia is measured. Why? Is it correct to name this “sleep disorders” then? Or is the insomnia scale used a measure of sleep disorders in general? The authors may explain why they have chosen to measure insomnia and not other sleep disorders, or frame it differently. 

In the revised version of the manuscript, we replaced ‘sleep disorders’ with ‘insomnia’ throughout the text, changing appropriately the article’s title as well.

[10]. Methods: Are the instruments used validated in previous research? Which language are they used in and are these versions validated? What are the Cronbach alphas for the scales? 

Validated Greek versions of instruments were utilized in the study. Cronbach’s alpha for the scales are included in the revised version of the manuscript.

[11]. The AIS description, line 94-95: The authors write that the instrument is designed for quantifying sleep difficulty based on the ICD-10 criteria. The ICD-10 criteria for what? Insomnia? 

The instrument is designed to quantify sleep difficulty based on the International Classification of Diseases 10th revision (ICD-10) criteria for insomnia. This is clarified in the revised version of the submission.

[12]. Line 123-129: Is this a text about the requirements of the methods section in the journal? Should be removed? 

The reviewer is right; this was an irrelevant intrusion. It has been removed from the revised version of the paper.

[13]. It would also be nice with a little more information about how the statistical analyses were performed, what specific part of the study aims was investigated with for example the t-test, and what with the regression? 

We revised the statistical analysis section, including more details to better address our hypothesis. The text has been modified as follows: “The independent groups t-test was performed to evaluate differences on continuous variables as to gender. The one-sample t-test was employed to determine differences on scores between nursing staff and the Greek general population. One-way ANOVA was used to test for differences on scores among included participants from various hospital departments. Pearson's correlation and stepwise linear regression analysis were also used for the evaluation of continuous variables.

[14]. There are very many tables, and too much focus on gender in the beginning. This is not part of the study aims, why are there separat analyses looking at this? I would suggest considering collapsing table 1, 2 and 3 into one where you show Mean and SD in the first column after the variables (for non-categorical variables, show frequencies for categorical variables as gender), then show the correlations - remove the n and the sig rows that now are in the correlation table. In that way it is easier to read the correlations. 

In the revised version of the manuscript we collapsed tables 1 & 2. We could not add-on to a single table the correlations’ results, but after removing the n and the significance rows, as advised, the correlations can be read more easily.

[15]. line 132: What is “working data”?

Thank you for your remark, it was wrong phrasing. We meant work experience (in years).

[16]. Part of the aim is to look at prevalence of sleep problems. Do you have any data from other samples/the same/similar samples at a time when there is no pandemic, to compare the % with? That 4.8% of the participants reported high levels of perceived stress does not give much information without anything to compare it with. A prevalence of 49.7% for insomnia sounds high, but how is this compared to before COVID-19, or compared to others? This is mentioned in the beginning of the discussion, but would it be possible to get these data from the general population or other studies to analyse the differences? 

Following the Reviewer’s suggestion, we included studies from the general population during the pandemic:

25.Voitsidis P., Gliatas I., Bairachtari V., Papadopoulou K., Papageorgiou G., Parlapani E., Syngelakis M., Holeva V., Diakogiannis I. Insomnia during the COVID-19 pandemic in a Greek population. Psychiatry Research 2020, 289, 113076.

References on insomnia before the pandemic crisis were included. In these studies, insomnia seems to be prevalent among nurses in Greek public hospitals as well as in hospitals in other countries:

26.Zhang C., Yang L., Liu S., Ma S., Wang Y., Cai Z., Du H., Li R., Kang L., Su M. et al. Survey of Insomnia and Related Social Psychological Factors Among Medical Staff Involved in the 2019 Novel Coronavirus Disease Outbreak. Frontiers in Psychiatry. 2020, 11, 306.

27.Qi J., Xu J., Li B.Z., Huang J.S., Yang Y., Zhang Z.T., Yao D.A., Liu Q.H., Jia M., Gong D.K., at al.  The Evaluation of Sleep Disturbances for Chinese Frontline Medical Workers under the Outbreak of COVID-19. Sleep Medicine 2020, 72, 1-4.

28.Kousloglou SA, Mouzas OD, Bonotis K, et al. Insomnia and burnout in Greek Nurses. Hippokratia 2014, 18, 150-155.

29.Leyva-Vela B, Jesús Llorente-Cantarero F, Henarejos-Alarcón S, et al. Psychosocial and physiological risks of shift work in nurses: a cross-sectional study. Cent Eur J Public Health 2018; 26: 183–189.

30.Infante-Rivard C, Dumont M, Montplaisir J. Sleep disorder symptoms among nurses and nursing aides. Int Arch Occup Environ Health. 1989; 61: 353-358.

Also, even before the pandemic crisis, moderate to high levels of stress were noted among nurses:

31.Adriaenssens J, De Gucht V, & Maes S. Causes and consequences of occupational stress in emergency nurses, a longitudinal study. Journal of Nursing Management,2015;23(3), 346-358.

32.Masa’Deh R, Alhalaiqa F, AbuRuz ME, et al. Perceived Stress in Nurses: A Comparative Study. Global Journal of Health Science; 2017; 9, 195-203.

33.Kshetrimayum N, Bennadi D, Siluvai S. Stress among staff nurses: A hospital based study. J Nat Sci Med, 2019;2:95-100.

[17]. Correlation analysis: It would be easier for the reader to read this table if the dependent variables (in the regression analysis) were presented at the bottom of the table, and that all the tables had the same order (demographic variables first, then FS, then PSS/AIS).

In the revised version of the manuscript we addressed these issues in tables 3 & 4.

[18]. The regression analysis: I am not surprised that family support was not significantly related to AIS when controlling for PSS. The correlation between PSS and AIS is high. Were all the assumptions for performing a regression analysis met in the data? 

Before performing stepwise multiple regression analysis we checked all the variables for the assumption of homoscedasticity (with inspection of scatterplots), and for the absence of multicollinearity (variance inflation factor <5).

[19]. It is interesting that family support is related to stress, but not to insomnia. It would be interesting to see if the results changed if the Stress analysis included the same variables as the Insomnia analysis (that is, also including age in the analysis for stress even though it was not correlated to stress in the correlation analysis - there are different opinions as to including variables with low/no correlation in a regression or not).

We conducted the suggested model for stress analysis, but the results did not change. We added the phrase: “…whereas age, and work experience were not significant (p >0.05, Table 4).”

[20]. line 170 “the independent variables that showed significant associations” - is this in the correlation analysis?

We changed the phrasing, to avoid possible misunderstanding, as follows: “On the basis of the results of the bivariate analyses, a stepwise multiple regression test was performed to identify the best predictors of the Perceived Stress Scale from the independent variables that correlated significantly with it in the correlation analysis (Family Support Scale and Athens Insomnia Scale scores) and to examine their contribution to this dependent variable’s variation (expressed as R Square). The final regression model showed that scores on the Athens Insomnia Scale and scores on the Family Support Scale were significant predictors of the dependent variable explaining 43.7% and 9,2% of the variance, respectively (F1,116=65.040, p=0.000, Table 4)….”

[21]. Discussion: Line 189: “Literature suggests” What literature? What is the reference? 

In the revised version of the manuscript we added the relevant reference:

34.Lai J, Ma S, Wang Y et al. Factors Associated With Mental Health Outcomes among Health Care Workers Exposed to Coronavirus Disease 2019. JAMA Network Open. 2020;3: e203976

[22]. There need to be a higher awareness of the causality of the variables (or lack of due to the cross-sectional design) in the discussion in general, especially when discussing the results from the regression. 

We tried to rephrase the concluding paragraphs in order to avoid causal assumptions. The phrase in line 199 "This cross-sectional study revealed a high prevalence of sleep disorders .." was changed to "This cross-sectional study suggested a high prevalence of sleep disorders .."

[23]. line 204, sentence reads: “there is evidence of a bidirectional association between insomnia and loneliness [28], providing an explanation for the positive effects of family support.” Which effects of family support are the authors talking about here? In the regression family support was not significantly related to insomnia. This is also relevant for the last sentence in the conclusion. 

We changed the phrasing, to avoid possible misunderstanding, as follows: “From another point of view, there is evidence of a bidirectional association between insomnia and loneliness [39], providing a different approach for the positive influence of family support.”

[24]. The implications of the main findings could be communicated more clearly. What is the most important finding and how can this improve the sleep/work of nurses during the pandemic? I think it is interesting that family support is related to stress, but not to insomnia. Why is this? 

The findings from this study showed that family support of the nursing staff did not directly influence rates of insomnia, but followed an indirect pathway via stress. More in detail, family support has a protective role on nurses' sleep by having a positive contribution on reducing stress levels; however if this increased sense of family support is temporary, it is not sufficient. In this context, it is highly recommended that nurse managers and policy makers pay more attention to this phenomenon. Early, targeted interventions should be considered that will include a network of specialized mental healthcare services to address psychological care needs and attenuate the possibility of escalating complications. Hospital administrators should be aware of the extent and sources of stress and psychological distress among frontline healthcare workers during disease outbreaks and implement interventions at the management level that will alleviate the impact of psychological pressure.

Again, we appreciate our reviewers’ effort and their useful recommendations on our paper and we hope that the revised version of our manuscript taking into account the changes requested, has a chance of subsequent acceptance for publication.
